



# Angle-domain common-image gathers from Fresnel volume migration

Tomi Jusri[1], Stefan Buske[1], Olaf Hellwig[1], and Felix Hloušek[1]

[1]Institute of Geophysics and Geoinformatics, TU Bergakademie Freiberg, Gustav-Zeuner-Str. 12, 09596 Freiberg, Germany

**Correspondence:** Tomi Jusri (tomi.jusri@geophysik.tu-freiberg.de)

**Abstract.** In complex geological settings, such as in hard-rock environments, Fresnel volume migration (FVM) has been successfully applied and found to deliver superior image quality compared to conventional imaging techniques. However, previous studies on FVM have mainly focused on obtaining kinematic seismic images, and the analysis of the migrated amplitudes has not received major attention. Therefore this study presents a method for constructing angle-domain common-image gathers

(ADCIGs) and common-angle stacks from FVM, which can facilitate prestack amplitude analysis from the migrated seismic data in the angle- domain. These ADCIGs were constructed inside the migration loops using phase slowness vectors derived from traveltime gradient fields. We then tested this method on synthetic and field seismic data and investigated the reliability of the output for amplitude versus angle (AVA) analysis. The test results obtained showed that the AVA responses from the common-angle stacks resemble that of the input synthetic shot gather of migration relatively well, indicating the

promising feasibility of AVA analysis from common-angle stacks. When implemented on field data acquired from a hard-rock environment, the proposed method can provide common-angle stacks with a higher signal-to-noise ratio and better reflection coherency compared to the common-angle stacks from the standard Kirchhoff prestack depth migration. This study extends the implementation of FVM toward amplitude analysis, which can help improve the feasibility of hard-rock characterization.

## 1 Introduction

Fresnel volume migration (FVM), as an extension of Kirchhoff prestack depth migration (KPSDM), is an effective seismic imaging technique that is especially used in the case of challenging geological environments, where conventional seismic imaging techniques mostly fail (Heinonen et al., 2019; Jusri et al., 2019; Singh et al., 2019). FVM has been applied in many geological settings dominated by crystalline rocks, such as in studies on deep drilling in Germany (Hloušek et al., 2015a, b; Hloušek and Buske, 2016), deep geothermal exploration in metamorphic rocks in Italy (Riedel et al., 2015; Jusri et al., 2019),

and mineral exploration in ultramafic rocks in Finland (Heinonen et al., 2019; Singh et al., 2019). FVM is also considered a robust imaging technique suitable for a broad range of exploration scales, including imaging around boreholes (Lüth et al., 2005), to investigate the subducted oceanic crust at the Chilean continental margin (Sick, 2006; Buske et al., 2009).

      However, previous studies on FVM have mainly focused on obtaining high-quality kinematic seismic images, and the amplitude-related aspects of this technique have not been thoroughly investigated. Understanding seismic amplitudes is essen-

tial because they are the link between kinematic seismic images and information on the rock-physical properties of the imaged



subsurface structures. One proper method to examine seismic amplitudes, especially for estimating rock-physical properties, is to perform an amplitude versus angle (AVA) analysis (Castagna, 1993; Castagna and Smith, 1994; Castagna et al., 1998). For this analysis, proper angle-domain common-image gathers (ADCIGs) must first be obtained.

ADCIGs can be extracted from ray- and wave-based migrations in a similar way, since they both characterize angle-dependent amplitude variations in the imaging domain (Baina et al., 2002). However, ray-based migrations such as KPSDM tend to be more efficient for large-scale 3D data sets and are favorable for migration velocity analysis in anisotropic media (Liu et al., 2015, 2018). Other studies have also indicated that KPSDM is a better choice for AVA analysis than more sophisticated migration techniques, such as finite-difference migration or reverse-time migration (Ehinger et al., 1996; O'Brien et al., 2019). ADCIGs can also be constructed after ray-based migrations, which usually produce migrated seismic data in offset-domain

common-image gathers (ODCIGs), followed by their transformation to ADCIGs (Ostrander, 1984; Smith and Gidlow, 1987; Sava and Fomel, 2003). However, some studies have also suggested that reliable ADCIGs can only be constructed during migration, which implies that ADCIGs should be calculated inside migration loops where migrated images are directly obtained for each angle range (Xu et al., 2001; Baina et al., 2002). In particular, Xu et al. (2001) presented a 2D Kirchhoff migration formula as an integral over all migration dip angles at an image point for each opening angle between rays from a source and

a receiver to a subsurface image point. Similar approaches to Xu et al. (2001) were proposed by Bleistein and Gray (2002) for a 3D data set, as well as by Brandsberg-Dahl et al. (2003), who mainly focused on the implementation of the method for AVA analysis. As an alternative to Xu et al. (2001), Liu et al. (2015, 2018) utilized the dynamic programming method (Wang et al., 1999) to calculate traveltimes in 3D transversely isotropic media, followed by ADCIG construction through KPSDM using traveltime gradient fields. Furthermore, Liu et al. (2015, 2018) calculated the scattering opening angles replacing surface-

related offsets by using the geometric relationship between the source and receiver phase slowness vectors (Audebert et al., 2002; Cheng et al., 2011). Liu et al. (2015, 2018) argued that the constructed ADCIGs are more accurate than those from the ODCIG-to-ADCIG transformation methods and help avoid the complexity that may arise with the common-angle migration proposed by Xu et al. (2001).

      The goals of this study are twofold: (1) to propose a method for obtaining ADCIGs from FVM, which can facilitate prestack

amplitude analysis from the migrated seismic data, and (2) to investigate the feasibility of AVA analysis from its output. We constructed ADCIGs inside the migration using phase slowness vectors derived from traveltime gradient fields. In contrast to the method of Liu et al. (2018), we calculated the traveltime gradient fields using a finite-difference solution of the eikonal equation (Podvin and Lecomte, 1991). To maintain compatibility for the AVA analysis, we implemented the proposed method using the exact Kirchhoff weighting function, which differs from the approximated versions implemented in previous studies

with FVM. We then tested the proposed method on synthetic and field seismic data sets, and we discuss its advantages and potential limitations.





## 2 Methodology

### 2.1 Fresnel Volume Migration

Using the exact Kirchhoff weighting function, FVM obtains the image value $U$ of a subsurface image point at the lateral
distance $x$ and depth $z$ by weighted summation of the recorded wavefield over the corresponding diffraction surface $A$ for this
image point (Buske et al., 2009):

$$U(x,z,t_{\mathrm{s}}(x,z)) = \iint\limits_{A} \frac{\partial U(x',z=0,t')}{\partial t} W(\zeta,\tau)\, f(x,z)\, dt'\, dx', \tag{1}$$

where:

$$W(\zeta,\tau) = -\frac{z\tau H(\tau - t_r)}{\pi r^2 \sqrt{\tau^2 - t_r^2}}, \tag{2}$$

$$\zeta = x' - x, \tag{3}$$

$$\tau = t' - t_s, \tag{4}$$

$$f(x,z) = \begin{cases} 1, & \text{if } d \leq r_f, \\ 0, & \text{if } d \geq 2r_f, \\ 1 - \dfrac{d - r_f}{r_f}, & \text{if } r_f < d < 2r_f. \end{cases} \tag{5}$$

Equation 1 considers the time derivative of the input seismogram $t'$ and a Heaviside function $H$ in the Kirchhoff weighting
function $W$. In this equation, $\zeta$, $z$ and $r$ represent the lateral distance of a point on the diffraction surface to the surface
peak, the depth of the image point, and the distance of the image point to the receiver, respectively. In addition, $t_r$ and $t_s$
represent the traveltimes from the image point to the receiver and the source, respectively. With the weighting function $f(x,z)$,
FVM modifies the standard KPSDM by limiting the back propagation of the recorded wavefield to the physically relevant
parts along the reflectors, based on the orthogonal distance from the image point to the ray $d$ and the radius of the Fresnel
zone $r_f$. Therefore, FVM suppresses the artifacts produced by smearing the amplitudes along the two-way traveltime (TWT)
isochrons and far from the actual reflectors, which commonly appear on the results from the standard KPSDM. As an example,
Fig. 1 shows a comparison between KPSDM and FVM output for the same migration parameters and synthetic input data.
The geologic model consists of one flat reflector lying at a depth of $2000\,\mathrm{m}$ in the medium. The figure shows that FVM limits
the TWT isochrons onto the relevant parts of the reflector and produces a relatively noise-free migration output compared to
KPSDM.

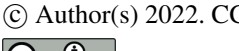



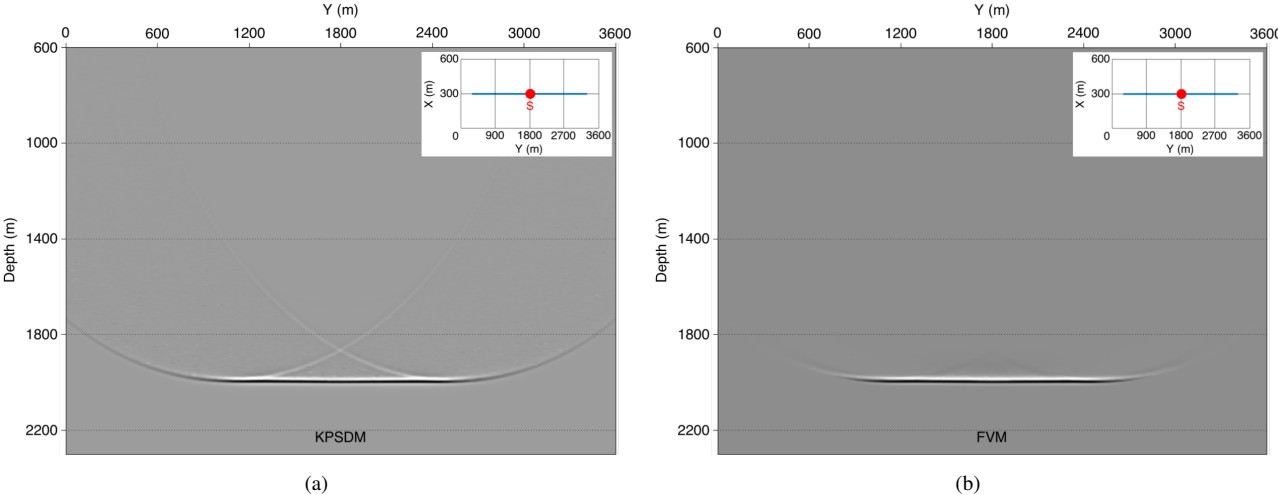

**Figure 1.** Comparison between (a) KPSDM and (b) FVM output for the same migration parameters and synthetic input data. The geologic model consists of one flat reflector lying at a depth of 2000 m in the medium. The red dot and blue line in the basemap mark the source and receiver positions, respectively.

## 2.2 Angle-domain common-image gather and common-angle stack

In prestack depth migration, a subsurface image point can be understood as the coincidence of the incident wavefront coming from the source with the emergent wavefront reflected at the image point and propagating toward the receiver (Audebert et al., 2002). These wavefronts can be parameterized by the source and receiver isochrons, wherein each isochron has its associated attributes, such as a traveltime gradient and a phase slowness vector. The resultant of the source and receiver phase slowness vectors produces an illumination slowness vector. Furthermore, four types of diffraction angles can be defined with the source and receiver phase slowness vectors (Audebert et al., 2002; Liu et al., 2018), as shown in Fig. 2. The four types of diffraction angles are:

1. scattering opening angle ($\alpha$), that is, the angle between the source ($\boldsymbol{P_s}$) and receiver ($\boldsymbol{P_r}$) phase slowness vectors;

2. scattering azimuth ($\phi$), that is, the azimuth of the plane containing both the source and receiver phase slowness vectors;

3. illumination dip angle ($\Theta$), that is, the angle between the normal and the illumination slowness vector ($\boldsymbol{P_m}$); and

4. illumination azimuth ($\Phi$), that is, the azimuth of the illumination slowness vector.

Fig. 2 shows that at the subsurface image point with a pair of source and receiver isochrons, the scattering opening angle $\alpha$ can be calculated from the phase slowness vectors $\boldsymbol{P_s}$ and $\boldsymbol{P_r}$ (Liu et al., 2018), and the incident angle $\theta$ can be calculated as half the scattering opening angle $\alpha$ as follows:


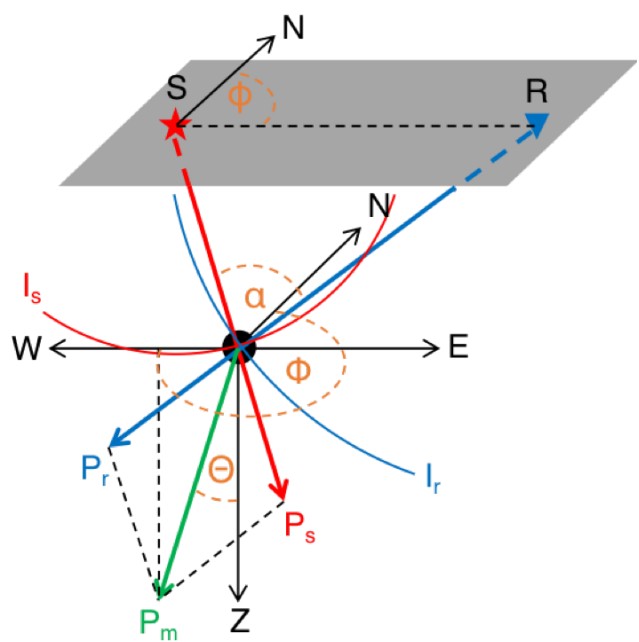

**Figure 2.** Geometric relationship between the phase slowness vectors and the diffraction angles (modified from Liu et al., 2018). In this figure, S and R represent the source and receiver on the surface, respectively; $I_s$ and $I_r$ represent the source and receiver isochrons, respectively; and N, E, W, and Z denote the spatial axes. The black dot at the center of the spatial axes marks the subsurface image point. In addition, $\boldsymbol{P_s}$, $\boldsymbol{P_r}$, and $\boldsymbol{P_m}$ represent the source phase, receiver phase, and illumination slowness vectors, respectively; whereas $\alpha$, $\phi$, $\Theta$, and $\Phi$ denote the scattering opening angle, scattering azimuth, illumination dip angle, and illumination azimuth, respectively.

$$\cos\alpha = \frac{\boldsymbol{P_s} \cdot \boldsymbol{P_r}}{|\boldsymbol{P_r}||\boldsymbol{P_r}|}, \tag{6}$$

$$\theta = \frac{1}{2}\left(\arccos\left(\frac{\boldsymbol{P_s} \cdot \boldsymbol{P_r}}{|\boldsymbol{P_s}||\boldsymbol{P_r}|}\right)\right). \tag{7}$$

Here, $\boldsymbol{P_s}$ and $\boldsymbol{P_r}$ can be determined from the source ($T_s$) and receiver ($T_r$) traveltime gradients:

$$\boldsymbol{P_s} = \left(\frac{\partial T_s}{\partial x}, \frac{\partial T_s}{\partial y}, \frac{\partial T_s}{\partial z}\right), \tag{8}$$

$$\boldsymbol{P_r} = \left(\frac{\partial T_r}{\partial x}, \frac{\partial T_r}{\partial y}, \frac{\partial T_r}{\partial z}\right). \tag{9}$$





An ADCIG at the subsurface image point can be constructed by binning the prestack images into the corresponding incident-angle bins $\theta$. Therefore, Eq. 1 can be reformulated as follows:

$$U(x, z, t_{\mathrm{s}}(x, z), \theta) = \iint\limits_{A} \frac{\partial U(x', z = 0, t', \theta)}{\partial t} W(\zeta, \tau) f(x, z) \, dt' \, dx'. \qquad (10)$$

Note that the implementation of Eq. 10 requires the binning interval and size to be properly chosen. The smaller the binning interval and size are, the more precise the final prestack images are in the incident-angle bins in representing the reflection
amplitude from a particular incident angle and the more memory resource (or longer runtime) needed for the computation. However, a very small binning interval and size will increase the risk of failing to capture the prestack images into the corresponding incident-angle bins. Hence, a binning interval and size of $1°$ and $0.5°$, respectively, are considered reasonable choices for most cases. These parameters will result in incident angles of $0° \leq \theta \leq 0.5°$ in the first bin, $0.5° < \theta \leq 1.5°$ in the second bin, $1.5° < \theta \leq 2.5°$ in the third bin, and so forth. Depending on the binning size, a correction factor needs to be multiplied to
the prestack images in the first and last incident-angle bins to compensate for the different sizes of these bins. For example, for a binning size of $0.5°$, the prestack images in the first and last incident-angle bins are multiplied by a factor of 2.

In practice, the final images from KPSDM and FVM are obtained by stacking the migrated amplitudes of the same image points from all the migrated shot gathers. With the constructed ADCIGs, the stacking can now be performed for different incident angles at every image point. The stacking can be viewed as filling the available incident-angle bins at every image point
with the corresponding migrated amplitudes from all the possible source-receiver offsets. The stacking results in a common-angle stack at each spatial location of the migrated seismic volume, which consists of optimal amplitude illumination and all the possible prestack images from all the possible incident angles at that location.

## 2.3 Amplitudes Versus Angle analysis

AVA analysis aims to examine the partitioning of incident wave energy at an interface formed by two media as a function of the
incident angle and the contrasts of the P-wave and S-wave velocities and the density between the two media. The partitioning of the wave energy in reflection seismic data can be described by angle-dependent reflection coefficients and formulated using the Zoeppritz equations (Zoeppritz, 1919). In this study, we performed an AVA analysis using the two-term Aki-Richard equations (Richards and Frasier, 1976; Aki and Richards, 1980) as an approximation to the exact solution of the Zoeppritz equations:


$$R(\theta) \approx A + B \sin^2 \theta, \tag{11}$$

$$A = \frac{1}{2}\left(\frac{\Delta V_P}{V_P} + \frac{\Delta \rho}{\rho}\right), \tag{12}$$

$$B = \frac{1}{2}\frac{\Delta V_P}{V_P} - 4\frac{V_S^2}{V_P^2}\frac{\Delta V_S}{V_S} - 2\frac{V_S^2}{V_P^2}\frac{\Delta \rho}{\rho}, \tag{13}$$

where $R$ denotes the reflection coefficient as a function of the incident angle $\theta$. $V_P$, $V_S$, and $\rho$ in the equations denote the averages of the P-wave velocity, S-wave velocity, and density of the two media, respectively. The operator $\Delta$ denotes the difference of the corresponding parameters between the media above and below the interface. The $A$ and $B$ are the AVA intercept and gradient, respectively. While the AVA intercept corresponds to the normal incidence reflectivity, the AVA gradient represents the amplitude changes with the incident angle. Equations 11–13 are considered a favorable approximation to the Zoeppritz equations for sufficiently small relative changes of elastic parameters on both sides of the interface. Furthermore, the equations are reliable for incident angles up to $30°$, and therefore, we show AVA responses only up to an incident angle of $30°$ in this study.

## 3  Synthetic data test

In general, the accuracy of estimating reflection coefficients from field data is restricted by the inaccurate knowledge of the source impulse and downward continuation process, resulting from unknown absorption effects and the true medium velocity (Temme, 1984). We can avoid these problems by adopting synthetic seismic data in which the medium velocity is known, and the source impulse can be accurately estimated.

### 3.1  Synthetic seismic data

We generated a synthetic shot gather for a model consisting of two isotropic and homogeneous layers, representing a shale overlaying a gas-sand lithology (Fig. 3). The reflector lies in the model at a depth of $2000\,$m, and the synthetic shot gather was generated using a finite-difference modeling code (Hellwig, 2017) with a zero-phase wavelet as the source signal. We performed the modeling for an off-end cable-spread acquisition geometry with 301 receivers, including a zero-offset receiver, a $5\,$s trace length, and $10\,$m source and receiver intervals. We then sorted these 301 synthetic traces to obtain a whole 2D seismic data set containing 301 synthetic shot gathers, in which each shot has a 3000 m far offset (Fig, 5a).

### 3.2  Zero-offset reflection coefficients

We began by analyzing the reliability of FVM and KPSDM using the exact Kirchhoff weighting function for calculating the zero-offset reflection coefficient. The zero-offset reflection coefficient $R$ at an image point $(x_0, z_0)$ from the FVM output can



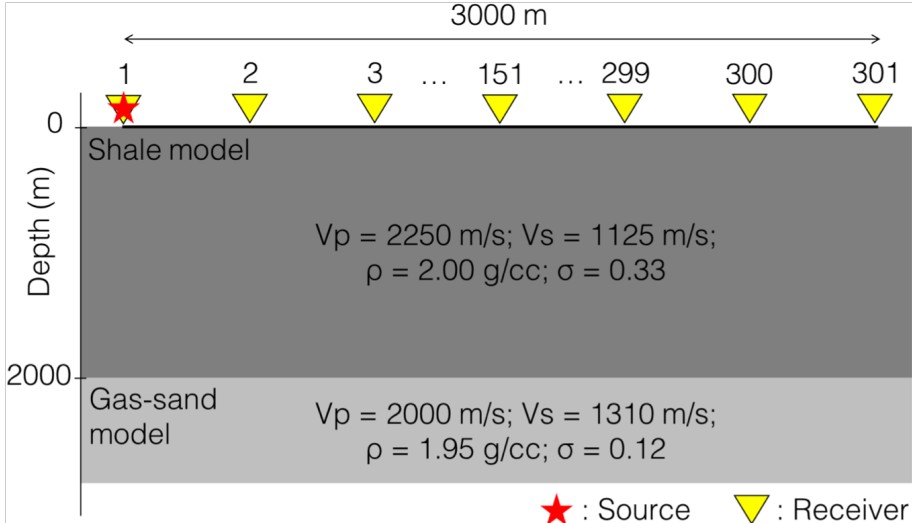

**Figure 3.** Geologic model for modeling the synthetic shot gather.

be obtained by dividing the amplitude of the migrated wavefield $U$ by that of the zero-offset incident wavefield $A$ at the image point (Temme, 1984):

$$R(x_0, z_0, \theta = 0°) = \frac{U(x_0, z_0, t_s(x_0, z_0))}{A(x_0, z_0, \theta = 0°)}. \tag{14}$$

     We obtained the migrated wavefield $U(x_0, z_0, t_s(x_0, z_0))$ by implementing FVM (Eq. 10) to the input synthetic shot gathers. The migration was conducted down to a depth of $4\,\text{km}$, with lateral migration grid intervals of $20$ and $200\,\text{m}$, parallel and

perpendicular to the 2D synthetic seismic line, respectively, and a vertical grid interval of $20\,\text{m}$. The true model velocity (Fig. 3) was used to compute the migration traveltimes. Figure 4a shows the output migrated shot gather at the middle of the seismic line.

     We calculated the amplitude of the zero-offset incident wavefield $A(x_0, z_0, \theta = 0°)$ at the image point by estimating beforehand the wavefield amplitude $A_0(\theta = 0°)$ at the source normal to the image point from the input synthetic shot gather data. For

a 2D isotropic and homogeneous medium, the amplitude decay as the wavefield travels from the source to a point at a distance of $x$ holds the following relationship (Aki and Richards, 1980):

$$A(x) = A_0 \cdot \frac{1}{\sqrt{x}}, \tag{15}$$

where $A(x)$ is the amplitude of the direct wave at a distance of $x$ from the source and $A_0$ is the wavefield amplitude at the source. We estimated $A_0$ by implementing Eq. 15 and utilizing the direct-wave amplitudes at the receivers. In particular, we





used the receiver at a relatively large distance from the source, i.e., at 1000 m, to obtain the direct-wave amplitude in the far-field of the source where the Eq. 15 holds. Finally, we calculated the amplitude of the zero-offset incident wavefield at the image point as follows:

$$A(x_0, z_0, \theta = 0°) = A_0(\theta = 0°) \cdot \frac{1}{\sqrt{r}}, \tag{16}$$

where $A_0(\theta = 0°)$ denotes the wavefield amplitude at the source normal to the image point, and $r$ is the distance from the

source to the image point. Figure 4b shows the calculated reflection coefficients from the migrated shot gather at the middle of the seismic line shown in Fig. 4a. Figure 4b also shows that the zero-offset reflection coefficient of the image point at the reflector below the source is $-0.068$ (rounded to three decimal places).

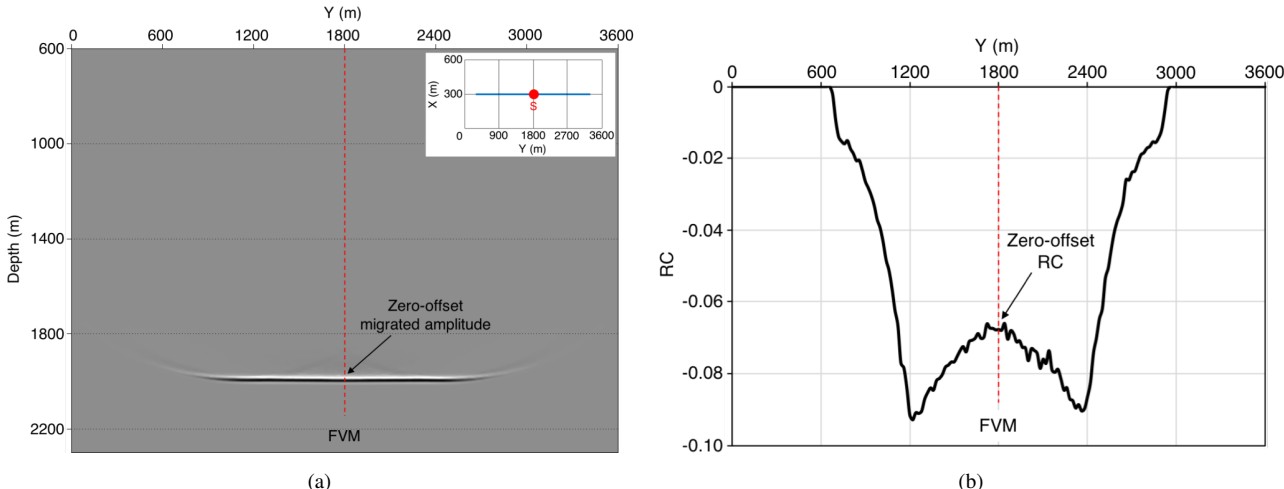

(a)    (b)

**Figure 4.** (a) FVM output from the geologic model in Fig. 3 and the shot gather at the middle of the seismic line. The red dot and blue line in the basemap mark the source and recording positions, respectively. (b) Reflection coefficients calculated from the migrated shot gather in (a) using Eq. 14. The red-dashed lines in both figures mark the zero-offset image point of the migrated shot gather. The zero-offset reflection coefficient (RC) is $-0.068$.

For comparison, we also obtained the zero-offset reflection coefficient from the input synthetic shot gather (Fig. 5a) and the theoretical zero-offset reflection coefficient directly from the geologic model in Fig. 3. Let $u_x$ be the reflected amplitude

recorded at the receiver with an offset of $x$ meters from the source and $A_0$ be the wavefield amplitude at the source. The reflection coefficient on the reflector at the midpoint between the source and the receiver $R_x$ is:

$$R_x = \frac{u_x}{A_0} \cdot \sqrt{2r}, \tag{17}$$





where $2r$ represents the distance from the source to the image point and then to the receiver. The reflection coefficients cal-
culated from the input synthetic shot gather are shown in Fig. 5b, where the reflection coefficient at the zero offset is $-0.068$

(rounded to three decimal places). The small spike observed on the reflection coefficient curve in Fig. 5b is due to a slightly
larger difference between the reflected amplitude at the zero and $10\,\mathrm{m}$ (the next receiver) offsets than the average differences
between the reflected amplitudes at the other consecutive offsets. In particular, the difference between the reflected amplitudes
at the zero and $10\,\mathrm{m}$ offsets is around $1.6\mathrm{e}{-}09$. On the other hand, the consecutive differences of the reflected amplitudes at the
further offsets (the differences between the 10 and $20\,\mathrm{m}$ offsets, the 20 and $30\,\mathrm{m}$ offsets, etc.) are around $1.8\mathrm{e}{-}12$, $1.6\mathrm{e}{-}12$,

and so forth. This peculiarity is mainly an artifact resulting from the forward modeling of the synthetic shot gather and is
unsubstantial.

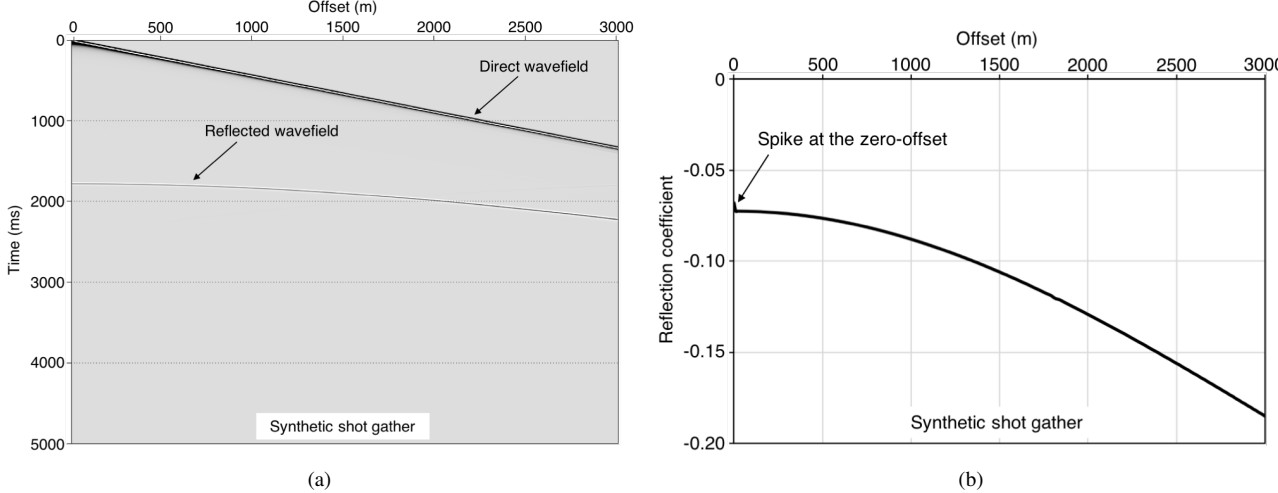

**Figure 5.** (a) Synthetic shot gather from the geologic model in Fig. 3. (b) Reflection coefficients calculated from the synthetic shot gather in
(a) using Eq. 17. The zero-offset reflection coefficient is $-0.068$. The spike on the reflection coefficient curve at the zero offset is explained
in the text.

In general, we can determine the theoretical zero-offset reflection coefficient using the impedance contrast between the two
layers, wherein the impedance is the product of the velocity and density of the medium. For a plane wave reflected at normal
incidence from a boundary surface of medium 1 lying on top of medium 2, we can calculate the reflection coefficient $R$ as

follows (Chopra and Castagna, 2014):

$$R = \frac{V_2\rho_2 - V_1\rho_1}{V_2\rho_2 + V_1\rho_1},  \tag{18}$$

where $V$ and $\rho$ are the velocity and density, respectively, with indices corresponding to the two media. With Eq. 18, the
theoretical zero-offset reflection coefficient was found to be $-0.071$ (rounded to three decimal places).



Overall, the results show that the zero-offset reflection coefficient calculated from the FVM ($-0.068$) is consistent with
that calculated from the input synthetic shot gather and different from the theoretical one by only around $0.003$, which is
reasonably negligible. This difference is mainly due to the numerical imprecision in the computation, especially given the fact
that we used a Real precision data format instead of Double Precision during our calculations. The results suggest that FVM
and KPSDM, along with the exact Kirchhoff weighting function, can handle the amplitudes of the input data very well. They
can also provide an accurate zero-offset reflection coefficient at an image point, provided that the amplitude of the incident
wavefield at the image point is accurately estimated, as in the case of synthetic seismic data.

### 3.3 Angle-domain common-image gather

We constructed ADCIGs from KPSDM and FVM using the synthetic seismic data with a binning interval of $1°$. Each bin
contains all the incident angles within $±0.5°$ from the corresponding bin center, except for the first and last bins, which contain
all the incident angles within $+0.5°$ and $-0.5°$ from the first and last bin centers, respectively. Figure 6 shows the ADCIGs
constructed from KPSDM and FVM at the image point on the reflector in the middle of the seismic line, from different shot
positions.

Figures 6a–6c also show that the reflections in the ADCIGs constructed from KPSDM are not focused on the actual reflec-
tor, which lies at a depth of $2000$ m. Migration artifacts are prominently present in those ADCIGs because of the smearing
amplitudes along the TWT isochron, and these artifacts become even more noticeable with the lateral distance between the
source and image point. On the other hand, Figs. 6d–6f show that FVM produces ADCIGs in which the reflections are focused
on the reflector as a result of limiting the TWT isochron at the relevant part of the reflector. This advantage causes the image
to be focused at the reflector in all ADCIGs from FVM, independently of the source locations.

To gain a better understanding of the ADCIG profiles, we consider the ADCIGs in Figs. 6c and 6f as an example and discuss
their construction through amplitude summation in the Kirchhoff integral of migrations. Figure 7a illustrates the amplitude
summations within KPSDM and FVM. The amplitude summation starts from the first receiver at a distance of $300$ m and
ends at the last receiver at a distance of $3300$ m along the Y-axis (see the basemap in Fig. 7a). These first and last receivers
correspond to the maximum incident-angle bin of $19°$ at both sides from the source position in the middle of the seismic line.
Notably, the amplitude summation of KPSDM starts with an extremely small magnitude, which is very close to zero at the first
receiver, and the summed amplitudes then start to grow toward the last receiver. Therefore, KPSDM produces amplitudes in
the entire incident-angle bins up to $19°$, as shown in Fig. 6c. On the other hand, the amplitude summation of FVM accumulates
significant amplitudes only up to incident-angle bins of $7°$, as can be observed in Fig. 6f. Furthermore, the curvature observed
from incident-angle bins of $0°$ up to nearly $4.5°$ in Fig. 7a corresponds to the part of the TWT isochron that coincides with
the diffraction surface, in which the migrated amplitudes are supposed to be weighted and summed up to produce an image.
Beyond this angle range, the curves are relatively flat, indicating very little or no contribution from the amplitudes beyond the
incident-angle bin of nearly $4.5°$ in the summation.

The following part focuses on AVA responses from ADCIGs in Figs. 6c and 6f. The AVA responses of the ADCIGs are
shown in Fig. 7b and plotted together with the theoretical AVA response and AVA response from the synthetic shot gather for





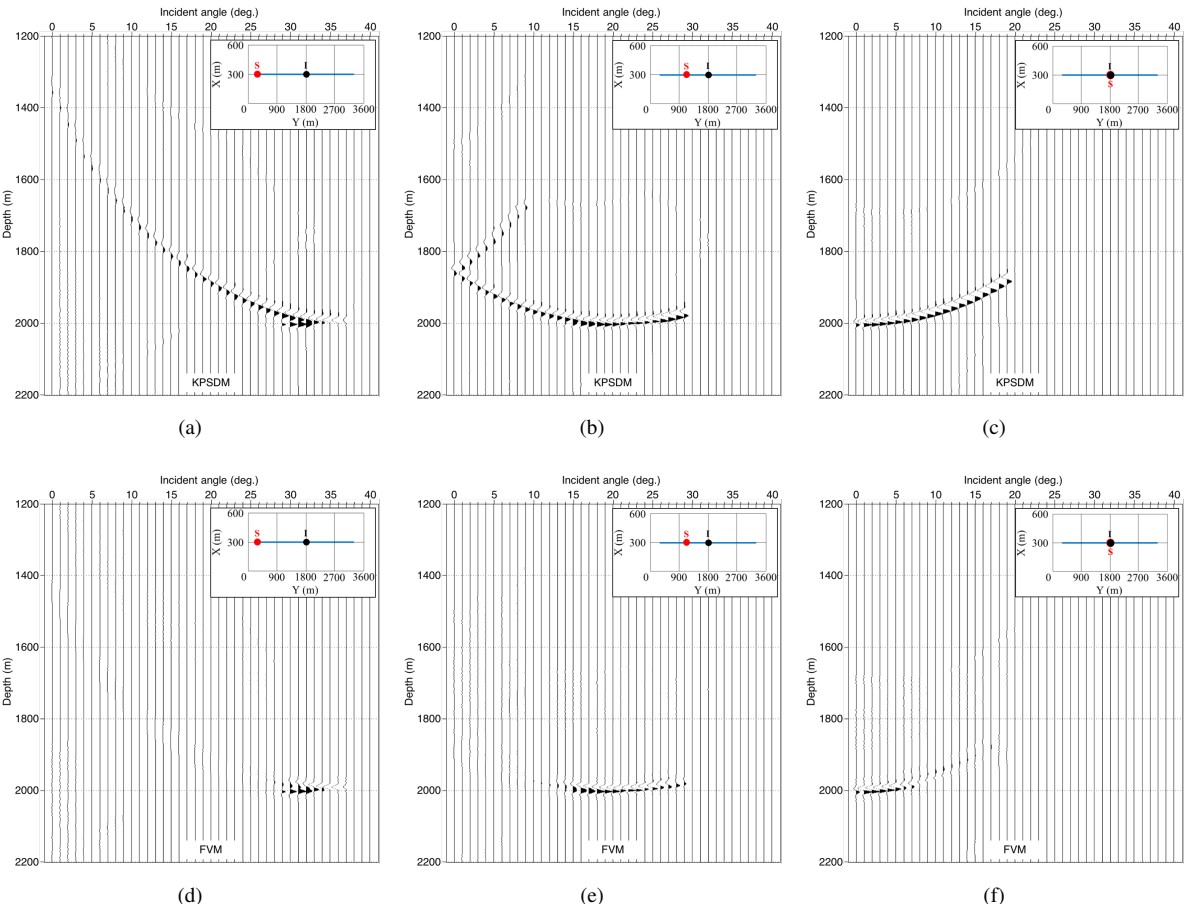

**Figure 6.** ADCIGs constructed from KPSDM (a–c) and FVM (d—f) at the image point on the reflector in the middle of the seismic line, from different shot positions. The blue line, red circle ("S"), and black circle ("I") in the basemaps represent the seismic line, shot position, and image point, respectively.

comparison. The intercept of each AVA response is normalized to the AVA intercept of the theoretical one. Differences between theoretical (exact) and seismic-based AVA responses for the same geologic model are common in practice, and Fig. 7b shows

that our case is no exception. The difference observed is mainly due to the deviation of the underlying conditions for the AVA response of the synthetic shot gather from the underlying conditions in the Zoeppritz equations, which are used for the theoretical AVA response. For example, the Zoeppritz equations imply that the amplitudes are equivalent to the reflection coefficients in the absence of modeling- and processing-related effects, such as transmission losses, attenuation, and divergence. On the other hand, seismic amplitudes are not directly proportional to reflection coefficients because of the effects mentioned

earlier. Certain distinguishing features of the Zoeppritz equations are further discussed at length by Allen and Peddy (1993), Castagna (1993) and Chopra and Castagna (2014). Nevertheless, for qualitative AVA analysis, Fig. 7b shows that the theoretical and AVA responses from the synthetic shot gather have similar values up to an incident angle of approximately 4.5°.

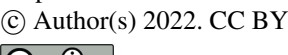



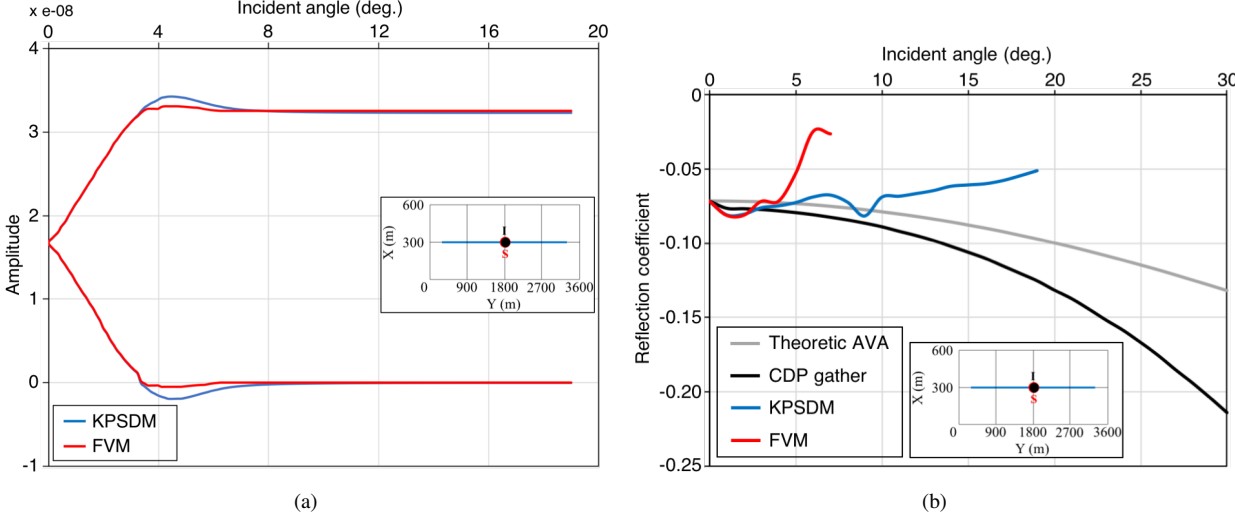

(a)                    (b)

**Figure 7.** (a) Amplitude summations at the image point in the ADCIGs in Fig. 6c and 6f, plotted as a function of incident angles. The blue line, red circle ("S"), and black circle ("I") in the corresponding basemap represent the seismic line, lateral shot position, and image point, respectively. (b) Comparison of the theoretical AVA response, AVA response from the synthetic shot gather (Fig. 4a), and AVA responses from the ADCIGs in Figs. 6c and 6f.

Figure 7b also shows that the AVA responses from the KPSDM and FVM relatively agree with the theoretical AVA response and AVA response from the synthetic shot gather up to an incident-angle bin of approximately $4°$. Beyond this incident-angle

bin, the AVA responses from the KPSDM and FVM are no longer comparable and significantly differ from the theoretical AVA response and AVA response from the synthetic shot gather. However, as mentioned earlier, Fig. 7a shows that the incident-angle bins up to approximately $4°$ correspond to a diffraction surface that truly contributes to the image. Therefore, the parts of the curves beyond the incident-angle bin of approximately $4°$ are no longer relevant for AVA analysis since they no longer contribute to the true image of the reflector. Furthermore, it is worth noting that up to the incident-angle bin of approximately

$4°$, the AVA responses from the KPSDM and FVM differ only slightly from the theoretical AVA response and the AVA response of the synthetic shot gather. For example, with an incident-angle bin of $1°$, from which the most prominent difference can be observed, the theoretical reflection coefficient, reflection coefficient from the synthetic shot gather, and reflection coefficient from both KPSDM and FVM are around $-0.072$, $-0.076$, and $-0.081$, respectively, which are relatively very close to each other.

**3.4 Common-angle stack**

Figures 8a–8f show common-angle stack profiles from KPSDM and FVM at different positions on the reflector. The figures show that with the common-angle stack becoming closer to the middle of the seismic line, more amplitudes contribute to higher incident-angle bins, and generally fewer migration artifacts are found in the profiles. In Fig. 8a, KPSDM produces only a small number of significant amplitudes for incident-angle bins of not more than $3°$ at the end of the seismic line. Figure



8a also shows prominent migration artifacts at larger incident-angle bins up to $17°$ due to smearing the amplitudes along the TWT isochrons. These artifacts are suppressed in the profile shown in Fig. 8b, mainly because of the destructive amplitude interference introduced by the stacking. Figure 8b also shows a higher contribution of amplitudes for incident-angle bins up to approximately $17°$. Beyond this angle, similar migration artifacts as in Fig. 8a dominate the profile up to an incident-angle bin of approximately $35°$. On the other hand, Fig. 8d–8e show that the migration artifacts from FVM are well suppressed

primarily by the focusing effect of FVM prior to the stacking. At the middle of the seismic line (Figs. 8c and 8f), the amplitude illumination within the incident-angle bins in the common-angle stacks is at the maximum value, and the profiles show the least migration artifacts from both migrations. Nevertheless, compared to KPSDM in Fig. 8c, the common-angle stack profile from FVM in Fig. 8f still shows fewer artifacts.

Figures 8g–8i show the AVA responses of reflections in the common-angle stacks plotted together with the theoretical AVA

response and the AVA response from the input synthetic shot gather for comparison. The intercept of each AVA response is normalized to the AVA intercept of the theoretical one. The figures also show that the AVA responses from the common-angle stacks become more correlated to the AVA responses from the synthetic shot gather as the common-angle stack becomes closer to the middle of the seismic line. Figure 8g shows that only a small part of the AVA responses from the KPSDM and FVM is correlated to the theoretical AVA response and the AVA response from the synthetic shot gather. This small part, which is

only up to incident-angle bins of approximately $3°$, is associated with amplitudes within the same range of incident-angle bins in Figs. 8a and 8d. Beyond this angle, the AVA responses are associated with the migration artifacts in Figs. 8a and 8d, and hence, are not relevant for the AVA analysis. Figure 8h shows that as the common-angle stack becomes closer to the middle of the seismic line, a larger part of the AVA responses from the KPSDM and FVM relatively correlate to the AVA response from the synthetic shot gather. Again, this correlated part, which is up to incident-angle bins of approximately $17°$, is associated

with amplitudes within the same range of incident-angle bins in Figs. 8b and 8e. Finally, at the middle of the seismic line, where the illumination of the amplitudes within the incident-angle bins in the common-angle stacks is at its maximum, the AVA responses from the common-angle stacks and synthetic shot gather are relatively well correlated.

## 4    Application to field data

We tested the proposed method on 2D seismic data acquired from an area in northern Finland that is dominated by ultramafic

rocks and known to host mineral resources. The seismic data acquisition was carried out under the project Experiment of Sodankylä Deep Exploration (XSODEX). Figure 9 shows the geometry of the seismic line, which has a total length of 22.26 km. The data acquisition was performed with a mixture of off-end and split-spread (roll-along) schemes, in which near and far offsets were set to 2 and 3300 m, respectively. A Vibroseis source was used, and the seismic signals were recorded with vertical-component geophones. The source and receiver intervals were set to 20 and 10 m, respectively. The data set consists of

452 shots with a maximum of 312 channels per shot, and 2001 samples per trace with a 2 ms sampling interval. Furthermore, we processed the seismic data with the primary aim of suppressing the noise and retaining the energy lost as a result of geometrical spreading. Table 1 summarizes the key processing components and parameters.





**Figure 8.** Common-angle stacks from KPSDM (a–c) and FVM (d–f) and the corresponding AVA curves (g–i) at the different positions of the image point on the reflector. The blue line, red circle ("S"), and black circle ("I") in the basemaps represent the seismic line, shot position, and image point, respectively.

We performed KPSDM and FVM on the processed seismic data in a 3D fashion within a rectangular area of $8 \times 22.5$ km (Fig. 9) and down to a depth of $8$ km. Through the migrations, we constructed ADCIGs up to an incident angle of $35°$, and





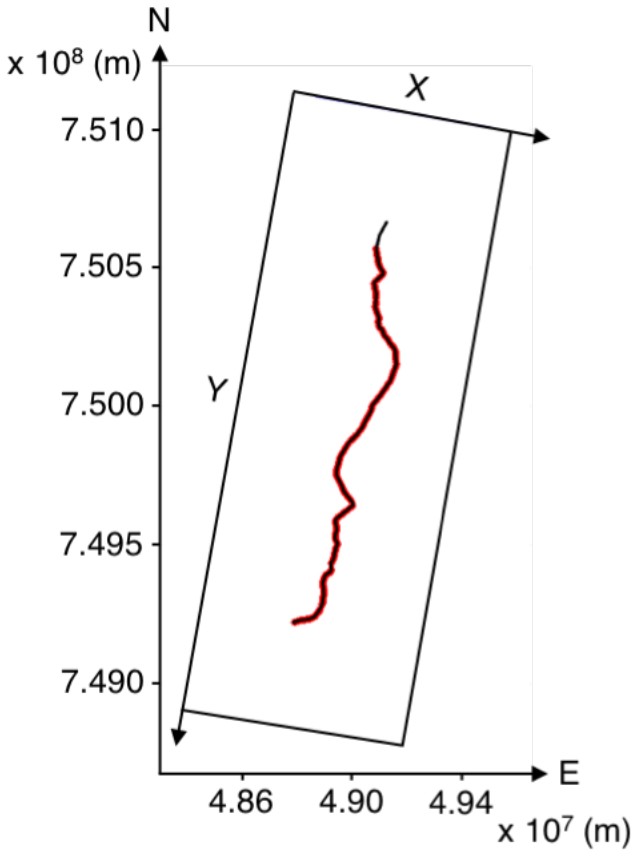

**Figure 9.** Geometry of the field seismic data. The red and black lines mark the source and receiver lateral positions, respectively. The migration was performed for the area within the blue box.

with a binning interval and binning radius of $1°$ and $0.5°$ from the corresponding bin center, respectively. A constant velocity of $5600$ m/s was used for the migration. Figure 10 shows the output migrated stack profiles. As a result of the focusing effect of the FVM, Fig. 10 shows that the migrated stack profile from FVM reveals noticeably more focused and coherent reflections a depth ranging from 10 to 18 km compared to KPSDM. On top of that, the FVM also reveals two distinct groups of coherent reflections at a depth ranging from 0 to 3 km between the distance ranges from 10 to 12 km and from 13 to 18 km, which cannot
be clearly distinguished in the profile from the KPSDM.

The red-dashed lines in Fig. 10 mark the position of the corresponding common-angle stacks shown in Fig. 11. Figure 11 also shows that the common-angle stack from FVM provides considerably less background noise and more coherent reflections than that from KPSDM. Furthermore, because of the lack of near-angle reflections, the nature of the field data does allow a proper AVA analysis. Nevertheless, the proposed method combined with the strength of FVM in image focusing has allowed us
to obtain the common-angle stack from the area, with considerably less background noise than the common-angle stack from KPSDM. The results indicate that when the characteristics of field data allow a proper AVA analysis, it is reasonable to expect



(a)

(b)

**Figure 10.** Migrated stack profiles from (a) KPSDM and (b) FVM. The red-dashed lines mark the positions of the corresponding common-angle stacks shown in Fig. 11. The blue-dashed line and red circle in the basemaps show the positions of the stacks and common-angle stacks, respectively, in Fig. 11.





**Table 1.** Key processing parameters for the field seismic data.

| Process | Key parameters |
|---|---|
| CDP binning | 5 m bin interval |
| Top-mute and noisy trace removal | Linear muting from 50 ms at zero offset to 750 ms at 3.3 km offset |
| Velocity analysis | 15 CDPs increment, 9 CDPs to combine |
| Spherical divergence correction | $t \times v(t)$ basis using picked velocity table, 0.25 dB/sec amplitude adjustment |
| Predictive deconvolution | 100 ms operator length, 16 ms prediction distance, 30–40–100–120 Hz bandpass-filter corners |
| Air-blast attenuation | 320 m/s energy velocity, 600 ms time gate width |
| Trace equalization | Maximum amplitude basis |
| Refraction statics | 250 m final datum elevation |
| Residual statics | Maximum power-autostatics method |

that FVM would provide better common-angle stacks with a higher signal-to-noise ratio and reflection coherency for the AVA analysis as compared to those from KPSDM. Further investigations with more suitable field data are necessary to confirm this premise.

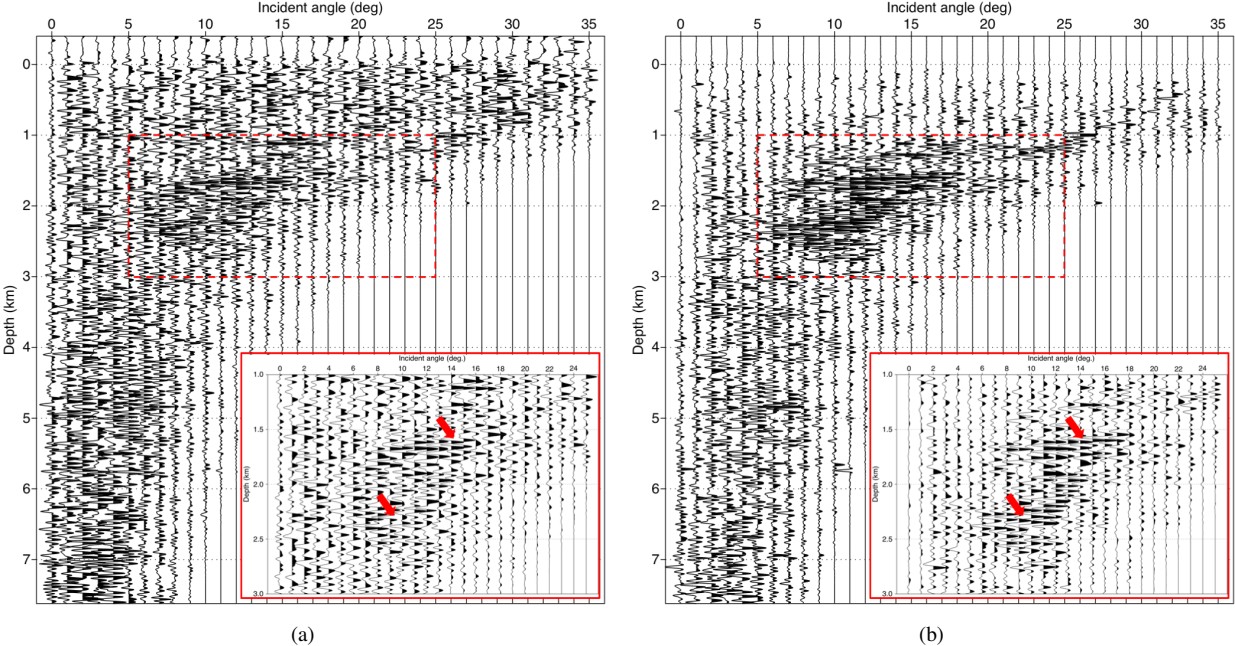

**Figure 11.** Common-angle stacks from (a) KPSDM and (b) FVM at the positions marked by the red-dashed lines in Figs. 10a and 10b, respectively. The red boxes show a magnified view of the areas marked by the corresponding red-dashed boxes in the figure. The red arrows highlight the enhancements in the profile from FVM compared to that from KPSDM.





## 5 Discussion

This study extends the implementation of FVM by introducing a method for obtaining ADCIGs and common-angle stacks from FVM, which facilitate prestack amplitude analysis from the migration output in the angle domain. The main advantages of this method are its practicality and compatibility with FVM codes. A vital component of this method is the calculation of the phase slowness vectors derived from the traveltime gradient fields. Since the traveltimes in FVM are calculated externally, the implementation of this method does not require a significant modification of the core architecture of the FVM codes. Furthermore, our test results show that the AVA responses of the common-angle stacks from FVM are relatively well correlated with the AVA response of the input synthetic shot gather of the migration, depending on the position of the common-angle stacks in the seismic line. This result indicates the promising feasibility of AVA analysis from the constructed common-angle stacks. Hence, the results obtained in this study can help improve the feasibility of rock characterization in challenging geological settings, such as in hard-rock environments.

We also observed that the reliability of the proposed method for AVA analysis is strongly affected by the amplitude summation in the Kirchhoff integral, the incident-angle binning, and the amplitude summation in the common-angle stacking. These aspects suggest that the proposed method neither preserves the true amplitudes nor provides the true reflection coefficients. Instead, it preserves the relative amplitudes in which an unknown multiplier may be present, but all the events are relatively correct (Hamspon, 2005). This finding agrees with the results of previous studies, which suggested that Kirchhoff migration properly treats the relative amplitudes of the input data (e.g., Resnick et al., 1987). Furthermore, in contrast to other studies, which claim to produce true-amplitude prestacks (e.g., Mosher et al., 1996; Baina et al., 2002), the proposed method provides common-angle stacks that are potentially compatible with qualitative AVA analysis only. However, in practice, qualitative AVA analysis is a reliable tool for detecting AVA anomalies (see, e.g., Ostrander, 1984; Smith and Gidlow, 1987; Rutherford and Williams, 1989) and has been used in the industry as a valuable tool for directly detecting hydrocarbons (Chopra and Castagna, 2014). On the other hand, quantitative AVA analysis relies primarily on deterministic corrections for true amplitudes, which can be rarely achieved in practice, and therefore, pose a great risk of seismic data misinterpretation (Chopra and Castagna, 2014). The misinterpretation risk is even more considerable when dealing with field data acquired from complex geological settings, such as hard-rock environments, which usually yield poor seismic data quality because of severe wavefield scattering.

Further studies are necessary to confirm the findings of this study. We suggest that further studies use synthetic data to probe the implementation of this method on 3D geometries and with extra-wide offsets. The latter will be especially useful in examining the feasibility of AVA analysis directly on ADCIGs instead of common-angle stacks. When field data is used, we recommend the inclusion of fully preserved amplitude processing prior to the migration, and we also recommend comparing the AVA response of the output with that from the same data after FVM. Moreover, the field data used should be suitable for standard AVA analysis, such as field data obtained from sedimentary environments. Once the implementation of the method on such field data is understood, we can extend the investigation to more challenging field data, such as field data acquired from hard-rock environments.





# 6    Conclusions

The method proposed in this study allows the construction of ADCIGs and common-angle stacks from FVM, which aids in

the prestack amplitude analysis from the migration output in the angle domain. Our test results indicate that the constructed common-angle stacks can be used for spotting AVA anomalies in qualitative AVA analysis. Hence, the results obtained in this study may eventually help improve the feasibility of rock characterization in challenging geological settings, such as in hard-rock environments.

*Data availability.*    The data that support the findings of this study are available on request from the corresponding author. The data are not

publicly available due to privacy or ethical restrictions.

*Author contributions.*    TJ and SB designed the study, developed the method and edited the manuscript. TJ wrote the codes, conducted the main tasks, such as seismic data processing and migration, and wrote the manuscript. OH provided the synthetic shot gather. TJ and FH designed the migration parameters for the field data. All authors contributed to discussions during the study and proofreading this manuscript.

*Competing interests.*    The authors declare that they have no conflict of interest.

*Acknowledgements.*    The field seismic data was acquired within the project XSODEX in cooperation with the Geological Survey of Finland (GTK). The time-domain preprocessing was carried out using ProMAX, thanks to the Halliburton-Landmark software grant. The migration codes were developed at the Institute of Geophysics Geoinformatics, TU Bergakademie Freiberg.



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
