# Peer review of "Angle-domain common-image gathers from Fresnel volume migration"

_Solid Earth, 2021_

## Author Comment (AC1)

*Red: Referee comments and questions*
Black: Authors responses

*The paper presents a method to generate angle-domain common-image gathers using Fresnel volume migration (FVM) and phase-slowness vectors obtained from traveltime gradient field, the latter used to bin the migrated data with the correct angles. The two methods are not new and have been published previously, but the combination of the two is new to the best of my knowledge. The paper is well-written and organized, and the figures are of good quality and support the text.*

1.  *Unfortunately, the real benefits and advantages of the proposed method over the classic Kirchhoff prestack depth migration (KPSDM) remain to be demonstrated.*

    See our response in number 4.

2.  *The synthetic and real data examples selected to illustrate the usefulness of the method to generate AVA-compliant gathers and thus AVA analysis are not convincing.*

    The main purpose of this study is to obtain angle-domain common-image gathers (ADCIGs) from FVM. We performed the AVA analysis mainly to describe or quantify the ADCIGs in terms of their AVA intercept and gradient attributes deduced from the AVA curves. In this context, the AVA analysis cannot be said "convincing" or "not convincing"—it was implemented simply by plotting the seismic amplitudes of the targeted reflector against the incident angles. Through the AVA analysis, we only show the seismic data amplitudes as they are. In another context, though, AVA analysis can also be referred to as some particular quantitative seismic interpretation (QSI) techniques which are often followed by inversion for rock-physical properties and take into account rock-physical information from borehole data. However, the implementation of such techniques would be a wholly independent project, which is not covered by the scope of the study presented here.

3.  *The synthetic model comprising only two layers is extremely simple.*

    See our response in number 5.

4.  *Obviously, there are merits in using simple models. They allow for a better understanding of the method and easier comparison with theoretical results (i.e. analytical AVA results). Unfortunately, the results and comparison with KPSDM results for such a simple model lead to the following question: why should anyone use FVM to generate angle-domain common-image gathers if results are almost identical to that of KPSDM (see figure 8 g, h, and i)? Whereas figure 8 might re-assure readers that the method provides as good results as KSPDM, it fails to demonstrate any advantages.*

    Previous studies show that KPSDM can well preserve the relative amplitudes of the input seismic data (e.g., Resnick et al., 1987). Figures 8g, 8h, and 8i imply that FVM can preserve the relative amplitudes of the input seismic data as good as KPSDM. The advantages of

FVM over the standard KPSDM for AVA analysis are noticeable when the migration is implemented on the field seismic data instead, in which a considerable amount of noise exists, such as the field data shown in Figure 11. Figure 11 clearly shows that FVM provides a common-angle stack with more coherent reflections and considerably less noise than that from KPSDM. This finding supports our main purpose in this manuscript.

However, we understand that this point may cause some misunderstanding, so we will make this point clearer in the revised manuscript.

4. *The problem is not the method but the simplistic model, which excludes the potential interference of smeared migration artifacts on AVA data. The impact of such interference on AVA curves (which should be more significant for KSPDM) can only be demonstrated by using a slightly more complex model with several geological layers. Such an example is necessary to show how FVM can help and improve AVA analysis.*

We use a simple geologic model for the following reasons:

a. As an initial study on this subject, i.e., amplitude investigation from FVM, we decided to start by using a simple geologic model to have full control over the expected results. We also expected reviewers to agree that such a methodological study should begin using a simple geologic model.
b. In hardrock environments, in which most FVM studies have been so far implemented, the velocity field tends to be relatively homogeneous. Therefore, we expect that a simple geologic model with a few layers and velocity contrasts can be sufficient to represent a typical geologic structure in a hardrock environment.
c. Investigation using a complex geologic model, such as multi-layer rock strata, would require a far more sophisticated approach, which takes into account various effects, including transmission losses at layer boundaries, variations of spherical divergence due to strong lateral velocity gradients, and even anisotropy. These effects play an important role in sedimentary environments but are less critical in hardrock environments. Therefore, we decided to stay away from such effects, which may mask the primary objective of our study and the ability to judge the successfulness of our approach.

We are currently still at an early stage in working on incorporating more advanced cases, such as anisotropy, into our migration algorithm. However, we strongly consider such investigation as a further study because it is another level that demands significantly more research, and it aims for other findings that are beyond the scope of the current study.

5. *The application of AVA analysis to hard rock environments is certainly interesting but also very challenging. The common-angle stacks shown in Figure 11 confirm this. As a reader, I wonder what useful AVA information can be extracted from those gathers. Assessing the added value of FVM over KPSDM for such data remains highly subjective. The FVM common-angle stack shown in the zoomed-in area of figure 8 looks more coherent but only over a limited range of angles. No reliable AVA analysis can be performed with this*

*data. So, how does it help demonstrate the usefulness of FVM for AVA analysis? Why even show it?*

AVA analysis can be considered a basic technique utilized by a broad spectrum of quantitative seismic interpretation (QSI) techniques. Your questions refer to QSI techniques based on AVA intercept and gradient attributes. The nature of the field seismic data may indeed not be suitable for such QSI techniques due to the lack of near-angle reflections. However, other QSI techniques do not necessarily require full-angle reflections; for example, elastic impedance inversion (Connolly, 1999), which mainly utilizes stacks at particular angle ranges. If suitable borehole data were available in the study area, the results from the field data shown in Figure 11 would allow us to perform elastic impedance inversion. Figure 11 also indicates that elastic impedance inversion would be more effective using the common-angle stack from FVM than KPSDM due to more coherent reflections and less background noise in the common-angle stack from FVM than that from KPSDM.

To make this point clearer, we will better explain the advantages of the results shown in Figure 11 in the revised manuscript.

6. *A field example from a less complex geological environment with supporting petrophysics (i.e. wireline logs for quantitative analysis) is needed. The authors propose this as future work. I would argue that this is needed in this paper. I recommend a major revision. This should provide enough time to include examples that can effectively help support the promising methodology presented in this paper.*

See our response in number 4.

7. *Minor question: The weights in equation 2 are a function of zeta and tau, but only tau is found on the term on the right-hand side. Am I missing something?*

The weighting function is parameterized by zeta implicitly. Zeta is the lateral distance between the receiver and the image point, i.e., the lateral distance of a point on the diffraction surface to the surface peak. This lateral distance is used to calculate the traveltimes from the source (ts) and receiver (tr) to the image point and to calculate tau.

We will add this explanation in the revised manuscript.

---

## Author Comment (AC2)

*Dear authors,*

*This manuscript considers the method of extracting the angle domain common image gather (ADCIG) and common angle stack (CAS) from Fresnel volume migration (FVM), where beside the kinematic properties of the migration image the focus is on improving the accuracy of the dynamic properties. The performance of the proposed method is investigated in amplitude versus angle (AVA) analysis. The manuscript is well written, organized satisfactorily and the idea is promising.*

*Main comments:*

*Fresnel volume migration is a well-developed method to modify the Kirchhoff pre-stack depth migration to eliminate the artifacts. On the other hand, the ADCIGs are the most precise gathers suggested as a solution for multi pathing, which are used for velocity and AVA analysis. There are some nice studies which show the superior performance of ADCIGs in imaging where the velocity model has complex structure.*

1. *Actually in front of complex geology, the single ray path assumption is violated and the multi-pathing occurs. In these situations the role of ADCIGs which uniquely define ray path based on their opening angle not their offset, becomes important. Therefore to show the predominance of ADCIG, authors need to use some geologically complex synthetic model, for example a model with some low velocity inclusion, or some benchmark model likes Marmousi to verify the dominant performance of ADCIGs constructed during FVM.*

We use a simple geologic model for the following reasons:

a. As an initial study on this subject, i.e., amplitude investigation from FVM, we decided to start by using a simple geologic model to have full control over the expected results. We also expected reviewers to agree that such a methodological study should begin using a simple geologic model.

b. In hardrock environments, in which most FVM studies have been so far implemented, the velocity field tends to be relatively homogeneous. In such an environment, velocity variations tend to be relatively not complex so that no significant multi-pathing occurs. Therefore, we expect that a simple geologic model with a few layers and velocity contrasts can be sufficient to represent a typical geologic structure in a hardrock environment.

c. Investigation using a complex geologic model, such as multi-layer rock strata, would require a far more sophisticated approach, which takes into account various effects, including transmission losses at layer boundaries, variations of spherical divergence due to strong lateral velocity gradients, and even anisotropy. These effects play an important role in sedimentary environments but are less critical in hardrock environments. Therefore, we decided to stay away from such effects, which may mask the primary objective of our study and the ability to judge the successfulness of our approach.

We are currently still at an early stage in working on incorporating more advanced cases, such as anisotropy, into our migration algorithm. However, we strongly consider such

investigation as a further study because it is another level that demands significantly more research, and it aims for other findings that are beyond the scope of the current study.

2. *To augment the manuscript to become easier to follow for the reader, I advise to add more explanation about the theory and the performance of FVM and ADCIG with some supporting figures in the theory section.*

It will be accommodated in the revised manuscript.

3. *In figures 1 and 4, and in all basemaps the horizontal label Y is meaning less and is introduced after using. Also in figure 11, it changed to X. So I advise to unify them and change it to distance, maybe become more sensible.*

It will be accommodated in the revised manuscript.

*Minor comments:*

4. *Increase the X and Y axis ticks and labels in figures 6, 7, 11. It is difficult to read them now.*

It will be accommodated in the revised manuscript.

5. *Line 68: Introduce the x' after equation 5 too.*

It will be accommodated in the revised manuscript.

6. *Line 78: Using the "noise-free" is an exaggerating phrase here, because beside the KPSDM result, it is a clear image but not generally free of any artifacts.*

It will be accommodated in the revised manuscript.

7. *Line 89 and 91: It is better to change one of the names for scattering azimuth angle or illumination azimuth angle, because their symbols in figure 2 are hardly distinguishable.*

It will be accommodated in the revised manuscript.

8. *Figure 8: There isn't any red circle in the figure which is introduced in the caption.*

It will be accommodated in the revised manuscript.

*Based on supplying a synthetic example which the single path is violated on, I recommend a major revision.*